

# Relationship between accelerometer-measured sleep duration and Stroop performance: a functional near-infrared spectroscopy study among young adults

Yanwei You[1,2,*], Jianxiu Liu[1,3,*], Xingtian Li[1,2], Peng Wang[1,2], Ruidong Liu[1,4] and Xindong Ma[1,5]

[1] Division of Sports Science & Physical Education, Tsinghua University, Beijing, China
[2] School of Social Sciences, Tsinghua University, Beijing, China
[3] Vanke School of Public Health, Tsinghua University, Beijing, China
[4] Sports Coaching College, Beijing Sport University, Beijing, China
[5] IDG/McGovern Institute for Brain Research, Tsinghua University, Beijing, China
* These authors contributed equally to this work.

Corresponding authors
Ruidong Liu, lrd5156@bsu.edu.cn
Xindong Ma,
maxd@mail.tsinghua.edu.cn

## ABSTRACT

**Objectives:** Short sleep is becoming more common in modern society. This study aimed to explore the relationship between accelerometer-measured sleep duration and cognitive performance among young adults as well as the underlying hemodynamic mechanisms.

**Methods:** A total of 58 participants were included in this study. Participants were asked to wear an ActiGraph GT3X+ accelerometer to identify their sleep duration for 7 consecutive days. Cognitive function was assessed by the Stroop test. Two conditions, including the congruent and incongruent Stroop, were set. In addition, stratified analyses were used to examine sensitivity. 24-channel functional near-infrared spectroscopy (fNIRS) equipment was applied to measure hemodynamic changes of the prefrontal cortex (PFC) during cognitive tasks.

**Results:** Results showed that sleep duration was positively associated with accuracy of the incongruent Stroop test (0.001 (0.000, 0.002), $p = 0.042$). Compared with the regular sleep ($\geq 7$ h) group, lower accuracy of the incongruent Stroop test ($-0.012$ ($-0.023$, $-0.002$), $p = 0.024$) was observed in the severe short sleep ($<6$ h). Moreover, a stratified analysis was conducted to examining gender, age, BMI, birthplace, and education's impact on sleep duration and the incongruent Stroop test accuracy, confirming a consistent correlation across all demographics. In the severe short sleep group, the activation of left middle frontal gyri and right dorsolateral superior frontal gyri were negatively associated with the cognitive performance.

**Conclusions:** This study emphasized the importance of maintaining enough sleep schedules in young college students from a fNIRS perspective. The findings of this study could potentially be used to guide sleep time in young adults and help them make sleep schemes.

## INTRODUCTION

Various evidence has supported the view that sleep serves an essential biological function for human health, especially mental and cognitive development (*Walker, 2009*; *Mason et al., 2021*; *Raven et al., 2018*). Young adults are increasingly prone to short sleep in today's society (*Owens, Adolescent Sleep Working Group & Committee on Adolescence, 2014*; *Shekari Soleimanloo et al., 2017*; *You et al., 2024*). In today's world, there are a multitude of causes leading to the fact that college students cannot achieve enough amount of sleep, including increased stress (*Kalmbach, Anderson & Drake, 2018*), more exposure to screens (*Whiting et al., 2021*), early start time for morning classes (*Boergers, Gable & Owens, 2014*), and delay of physiologic sleep time (*Gao, Terlizzese & Scullin, 2019*). Based on the National Sleep Foundation's guidelines for young adults, sleeping fewer than 7 h was classified as short-duration sleep (*Hirshkowitz et al., 2015*). Diverse ramifications of lacking sleep (<7 h per night) offer a strong support to the view that short sleep would cause sever physiological effects through its complexity.

Lack of sleep has been proven to be associated with a wide range of health consequences (*You et al., 2023f*; *Bertisch et al., 2018*; *You et al., 2023a*), including weakened cardiovascular circulation (*Penev, 2011*), elevated inflammatory response (*You et al., 2022*), impaired neurobehavioral functions such as thinking and memory (*Koa & Lo, 2021*). Moreover, prolonged wakefulness and short sleep can be detrimental to cognitive performance (*Wheaton & Claussen, 2021*). A recent meta-based study reported the correlation between sleep and cognitive function in school-aged students which showed that low sleep duration was correlated with poor cognitive and academic performance (*Astill et al., 2012*). A number of studies have identified a positive association between sleep duration and cognitive outcomes (*Koa & Lo, 2021*; *Yaffe et al., 2016*), which from another perspective, suggested that inadequate sleep might be linked to impairments in higher-level and complex cognitive abilities. Previously, studies explored possible neural mechanisms underlying the association between sleep and cognition in adolescents (*Telzer et al., 2015*; *Tarokh, Saletin & Carskadon, 2016*). EEG studies conducted on young individuals, measuring the electrical activity of the brain, provided convincing evidence that inadequate sleep can impact neural function. This is associated with deficiencies in memory formation, learning, executive functions, and emotional well-being (*Jan et al., 2010*). However, the perspective from hemodynamics deserves further study.

The noninvasive imaging of cortical activity provided by functional near-infrared spectroscopy (fNIRS) is becoming increasingly popular in numerous physiology or behavior studies (*You et al., 2023d*; *Herold et al., 2018*). This equipment holds an advantage over EEG due to its superior spatial resolution, allowing for more precise localization of neural activity. Additionally, fNIRS exhibits reduced sensitivity to artifacts from muscle movements, offering enhanced signal specificity compared to EEG. fNIRS served as an indicator of neuronal activity, which made it possible for the quantitative monitoring of

oxy-hemoglobin (oxyHb) and deoxy-hemoglobin (deoxyHb) changes by detecting variations from the source to the detector using near-infrared light wavelengths (*Pan, Borragan & Peigneux, 2019*). Several studies have used fNIRS to explore the relationship between brain activity and cognition (*You et al., 2023e*; *Pinti et al., 2020*; *Blanco et al., 2022*). For more mechanisms, neurovascular coupling (NVC), also known as "functional hyperemia," stands as a crucial homeostatic mechanism ensuring sufficient blood supply to the brain in times of heightened neuronal activity. Earlier research found that neurovascular coupling played a role in the cognitive process of acute post-sleep deprivation in human and mice (*Csipo et al., 2021*; *Tarantini et al., 2015*; *Tarantini et al., 2017*).

Prefrontal cortex (PFC), which is involved in executive functions, attentional control, and cognitive flexibility, was selected as the region of interest in this study. The PFC has been extensively implicated in higher-order cognitive processes, including response inhibition, working memory, and decision-making, which are central to Stroop test performance (*Miller, 2000*; *Friedman & Robbins, 2022*). The PFC plays a crucial role in various developmental aspects that unfold during adolescence, serving as a pivotal hub that connects cortical and limbic brain regions associated with cognitive functions (*Larsen & Luna, 2018*). Additionally, the PFC receives abundant neuromodulatory input from structures involved in sleep and arousal regulation, and compromised PFC function consistently manifests as a symptom in the sleep-deprived brain (*Goldstein & Walker, 2014*; *Krause et al., 2017*). This brain region is particularly sensitive to sleep-related changes due to its susceptibility to short-sleep's effects (*Curcio, Ferrara & De Gennaro, 2006*; *Anastasiades et al., 2022*), making it a prime candidate for investigating the potential neural underpinnings of the relationship between sleep duration and cognitive functioning.

Sensor-based accelerometers are common measures to assess sleep status. The rationale for using accelerometers to assess sleep lies in their capacity to objectively and non-invasively monitor movement patterns, providing valuable insights into sleep duration. This study was based on the premise that insufficient sleep may be associated with poorer cognitive function. We hypothesized that this correlation may involve the activation levels of critical brain regions within the PFC. Based on the current evidence, using neuroimaging measurement (fNIRS and sensor-based accelerometer), this study aimed to investigate (i) the relationship between sleep duration and cognitive performance in young college students; (ii) the association between sleep duration and hemodynamic activations.

## METHODS

### Participants

A total of 61 healthy right-handed college students in Beijing, China were initially recruited. The enrollment criteria for participants in this research encompassed the following aspects: (1) individuals being healthy young adults; (2) falling within the age range of 20 to 30 years, devoid of chronic ailments or mental disorders; (3) exhibiting right-handedness; (4) the absence of colorblindness or color weakness. This study was approved by the academic ethics committee of Tsinghua University (IRB 20190091) and

was in accordance with the Declaration of Helsinki (1964). All recruited subjects provided the written informed consent to participate in this study.

## Study design

This study was a cross-sectional study, which aimed at exploring the relationship between sleep duration and cognitive performance in young college students. Participants enrolled in this study were asked to wear an ActiGraph GT3X+ (Pensacola, FL, USA) accelerometer to identify their sleep duration for 7 consecutive days (covering both workdays and weekends). They wore an accelerometer on the wrist of their non-dominant hand except for showering and water-based activities (*Bingham et al., 2016*). To ensure data reliability, the collected accelerometer data were considered valid if they spanned a minimum of three days per week, comprising at least two workdays and one weekend day (*Migueles et al., 2017*; *Fairclough et al., 2017*). Utilizing the ActiLife software, the gathered data underwent analysis following the Actigraph guidelines. To obtain the daily sleep duration in minutes, the processed accelerometer data underwent a validated algorithm integrated into the software. Utilizing established sleep detection metrics, the algorithm incorporates movement patterns and periods of immobility to identify sleep episodes and compute daily sleep duration in minutes. A Stroop test was conducted. During the test, participants' cortex activations were examined using the fNIRS equipment.

## Cognitive Stroop test paradigm

In the Stroop test, participants were presented with a widely used cognitive assessment test aimed at measuring their selective attention and processing speed (*Ludyga et al., 2019*). As shown in Fig. S1, a computer-based version of the Stroop test was used to evaluate cognitive performance. Two conditions, including the congruent and incongruent Stroop, were set. Recruited subjects were asked to press arrow buttons on the keyboard to identify the color rather than the actual meaning of the word presented on the computer screen. For example, participants were shown a series of color words ("red," "blue," "green," *etc.*) displayed in a font color that did not match the word's actual meaning (*e.g.*, the word "red" displayed in blue font). The Stroop design was set by six test blocks containing 36 trials, with 30 s resting periods between each block. Before the formal test, two training blocks were provided for practice. The task was administered using E-Prime software, a well-established platform for creating and conducting psychological experiments, ensuring standardized presentation and response collection. Considering that morning assessments are known to reflect relatively stable cognitive performance and minimize the influence of diurnal variations, Stroop tests were administered in the morning.

## fNIRS measurements

The fNIRS equipment was used to acquire prefrontal hemodynamic activities during the Stroop task. A multi-channel, continuous wave, fNIRS imaging system (Oxymon MkIII and Octamon; Artinis Medical Systems, Utrecht, The Netherlands) with 24 channels and a sampling frequency of 10 Hz was applied. The wavelengths of light were 756 and 853 nm. Referring to *Vergotte et al. (2017)*, a spatial registration was used to estimate the cortical

regions covered by the fNIRS probe, and the positioning system was based on a 3D digital locator (Fastrack; Polhemus, Colchester, VT, USA) to obtain an estimation of the cortical regions covered by the fNIRS probe. Probe positions were set according to the international 10–20 system and a probe set of eight sources and 10 detectors with 3 cm distance was placed on the prefrontal cortex (PFC). Additionally, the digitized electrode position was automatically combined with MNI template brain (Fig. S2). The specific positions of reference electrodes were left and right earlobes, nasal root, parietal lobe and occipital eminence. Details of the spatial registration and the corresponding Broadmann atlas were provided in Table S1.

The fNIRS data were extracted and processed using Artinis software (Artinis Medical Systems) and NIRS-SPM (a package for statistical analysis of fNIRS signals based on SPM5, http://www.fil.ion.ucl.ac.uk/spm/). The wavelet MDL (minimum description length) method (Rorden & Brett, 2000) was applied to filter artefacts and noise such as Mayer waves. Referring to the previous study (Yennu et al., 2016), processed signals were then fitted into a generalized linear model using the hemodynamic response function, and the regression coefficient β value of the generalized linear model reflected the hemodynamic activation of each channel. BrainNet Viewer toolbox in Xia, Wang & He (2013) (The MathWorks, Inc., Natick, MA, USA) was used to visualize the activation of brain images.

## Statistics

Demographic characteristics were presented as mean ± standard deviation for continuous variables and numbers with percentages for category variables. A generalized linear regression model (GLM) was used to assess the correlation between sleep duration and cognitive test results (You et al., 2023c). GLMs are widely recognized and frequently employed as a fundamental analytical approach in the context of a conventional cross-sectional investigation. This method holds significant prominence owing to its versatile applicability across various scientific domains, encompassing disciplines such as epidemiology, social sciences, and health research. In addition, stratified analyses were used to examine sensitivity. Referring to the previous literature (Hirshkowitz et al., 2015), sleep duration was categorized into regular (≥7 h), light short (6.5–7 h), mild short (6–6.5 h) and severe short (<6 h) sleep. The birth place was categorized into cities and rural areas based on the location. According to the tri-sectional quantiles, age was categorized into <24, [24, 26), and ≥26 groups; BMI was categorized into <19.9, [19.9, 22.9), and ≥22.9 groups; taking the freshman year as 1, education year was categorized into <6, [6, 8), and ≥8 groups. All analyses were performed with the statistical software packages R (http://www.R-project.org, version 4.0.3) and MATLAB (2019b). $p$-value < 0.05 was considered to be statistically significant.

# RESULTS

## Demographic characteristics

The general demographic information of the participants is shown in Table 1. A total of 61 healthy right-handed college students in Beijing. A total of three participants dropped out

**Table 1 Demographic characteristics of participants.**

|  | Mean ± SD |
| --- | --- |
| Education (year) | 6.74 ± 1.86 |
| BMI (kg/m$^2$) | 21.80 ± 3.09 |
| Age (year) | 23.84 ± 5.04 |
| Accuracy (congruent Stroop) (%) | 0.99 ± 0.01 |
| Accuracy (incongruent Stroop) (%) | 0.98 ± 0.01 |
| Reaction time (congruent Stroop) (ms) | 0.64 ± 0.06 |
| Reaction time (incongruent Stroop) (ms) | 0.69 ± 0.07 |
| Sleep duration (min/day) | 373.10 ± 44.10 |
|  | N (%) |
| Birth place | |
| City | 41 (70.69%) |
| Rural | 17 (29.31%) |
| Gender | |
| Male | 25 (43.10%) |
| Female | 33 (56.90%) |
| Sleep duration (as category) | |
| Regular sleep duration (≥7 h) | 8 (13.79%) |
| Light short sleep (6.5–7 h) | 11 (18.97%) |
| Mild short sleep (6–6.5 h) | 18 (31.03%) |
| Severe short sleep (<6 h) | 21 (36.21%) |

of the study without taking the Stroop test. Finally, 58 participants with eligible data were used for the final analysis. Among them, 43% of them were males. The average age of them was 23.84 ± 5.04 years. More than half of them were born in cities. Participants in this study had an average sleep duration of 373 min per day, which was less than 7 h. 15% of participants got enough sleep (≥7 h) by WHO, and 36% of total participants got severe short sleep less than 6 h per night.

## Sleep duration and cognitive performance

Table 2 presents the correlation between sleep duration and Stroop performance. Initially, sleep duration was represented in a continuous form to explore the associations. β (95% CI) of Stroop test results were presented as study results. The congruent Stroop tests were found to be uncorrelated statistically. However, a significant association was identified in the accuracy of the incongruent Stroop test (0.001 (0.000, 0.002), $p = 0.042$). Moreover, the correlation between sleep duration (as a continuous variable) and reaction time of the congruent and incongruent Stroop tests was investigated (Table S2). Although a downward trend was found in the relationship between sleep duration and reaction time of the incongruent Stroop test (−0.001 (−0.005, 0.004)), no statistically significant associations were identified.

**Table 2** The associations between sleep duration with the congruent and incongruent Stroop test accuracy.

| | Accuracy of congruent Stroop test | | Accuracy of incongruent Stroop test | |
|---|---|---|---|---|
| | β (95% CI) | p-value | β (95% CI) | p-value |
| **Sleep duration as continuous variable** | | | | |
| Sleep (min/day) | 0 [−0.001 to 0.001] | 0.813 | 0.001 [0.000–0.002] | 0.042 |
| **Sleep duration as category variable** | | | | |
| Regular sleep duration | Reference | | Reference | |
| Light short sleep | 0.001 [−0.008 to 0.011] | 0.770 | −0.005 [−0.017 to 0.008] | 0.456 |
| Mild short sleep | 0.002 [−0.006 to 0.011] | 0.587 | −0.007 [−0.018 to 0.004] | 0.218 |
| Severe short sleep | 0.002 [−0.007 to 0.010] | 0.692 | −0.012 [−0.023 to −0.002] | 0.024 |

**Table 3** The associations between sleep duration with the congruent and incongruent Stroop test accuracy stratified by gender.

| | Accuracy of congruent Stroop test | | Accuracy of incongruent Stroop test | |
|---|---|---|---|---|
| | β (95% CI) | p-value | β (95% CI) | p-value |
| **Male** | | | | |
| Regular sleep duration | Reference | | Reference | |
| Light short sleep | −0.009 [−0.034 to 0.015] | 0.441 | 0 [−0.036 to 0.036] | 1.000 |
| Mild short sleep | −0.005 [−0.027 to 0.018] | 0.671 | −0.015 [−0.049 to 0.018] | 0.344 |
| Severe short sleep | −0.005 [−0.027 to 0.017] | 0.667 | −0.014 [−0.047 to 0.019] | 0.387 |
| **Female** | | | | |
| Regular sleep duration | Reference | | Reference | |
| Light short sleep | 0.004 [−0.007 to 0.015] | 0.477 | −0.008 [−0.020 to 0.004] | 0.171 |
| Mild short sleep | 0.003 [−0.007 to 0.014] | 0.529 | 0 [−0.011 to 0.011] | 1.000 |
| Severe short sleep | 0.002 [−0.009 to 0.012] | 0.736 | −0.014 [−0.025 to −0.003] | 0.015 |

Furthermore, sleep duration was categorized into four levels, with the regular sleep duration as the reference. There was a significant correlation between the Stroop test accuracy and the severe short sleep. Compared with the regular sleep population, lower accuracy of the incongruent Stroop test (−0.012 (−0.023, −0.002), $p = 0.024$) was observed in a population whose sleep was less than 6 h per night. When it comes to the reaction time (Table S2), compared with regular sleep group, decreasing sleep time at night was associated with longer reaction time in the incongruent Stroop test (for light short sleep group: 0.025 (−0.043, 0.093), for mild short sleep group: 0.013 (−0.048, 0.074), and for severe short sleep group: 0.014 (−0.046, 0.073)), however, such findings were not significant from a statistical perspective.

## Subgroup and stratified analysis
Given that gender differences produced varying effects on sleep status and cognitive function, the data were stratified by gender for further exploration (Table 3). Despite

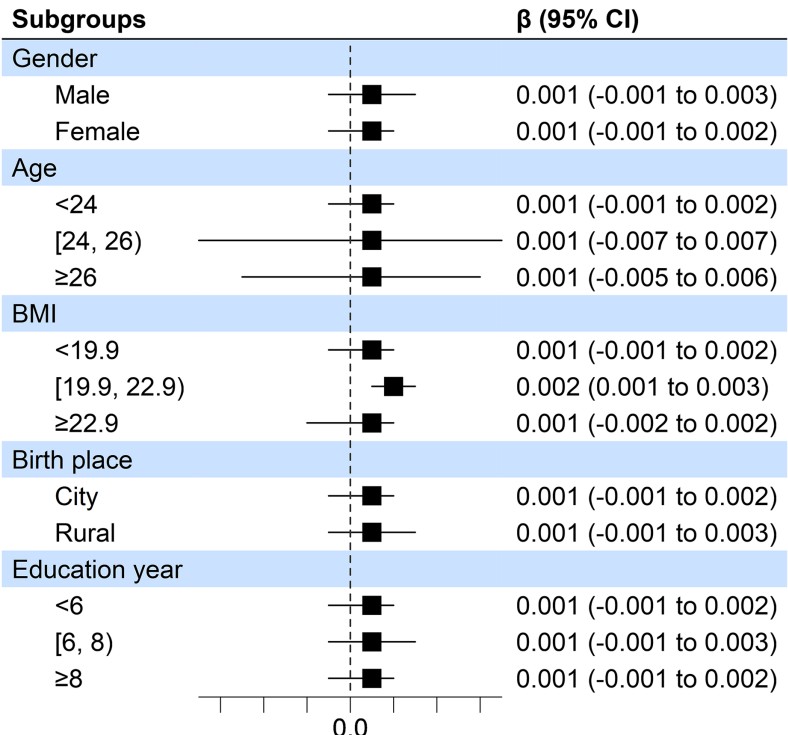

**Figure 1 Forest plots of associations between sleep duration and accuracy of the incongruent Stroop test.**

finding a negative trend that decreasing sleep duration was associated with lower accuracy in the incongruent Stroop test, no significant difference was found in the male group. On the other hand, for the female group, compared to those who slept over 7 h, significantly lower accuracy was observed for those who slept less than 6 h ($-0.014$ ($-0.025$, $-0.003$), $p = 0.015$) in the incongruent Stroop test.

Moreover, in consideration of the gender, age, BMI, birth place, as well as education year on the association between sleep duration and accuracy of the incongruent Stroop test. A stratified analysis was conducted for demographic factors of participants in this study (Fig. 1). No significant evidence was identified that demographic factors had an influence on findings mentioned above. This also confirmed the overall trend of positive correlation between sleep duration and accuracy of the incongruent Stroop test accuracy in all groups.

## fNIRS results

Based on the above results, it was found that sleep duration of college students was significantly associated with accuracy of the incongruent Stroop tests. Hence in this section, we focused on the correlation between measures of prefrontal cortex activation by hemodynamics using the oxyHb index and cognitive performance assessed by accuracy of the incongruent Stroop tests.

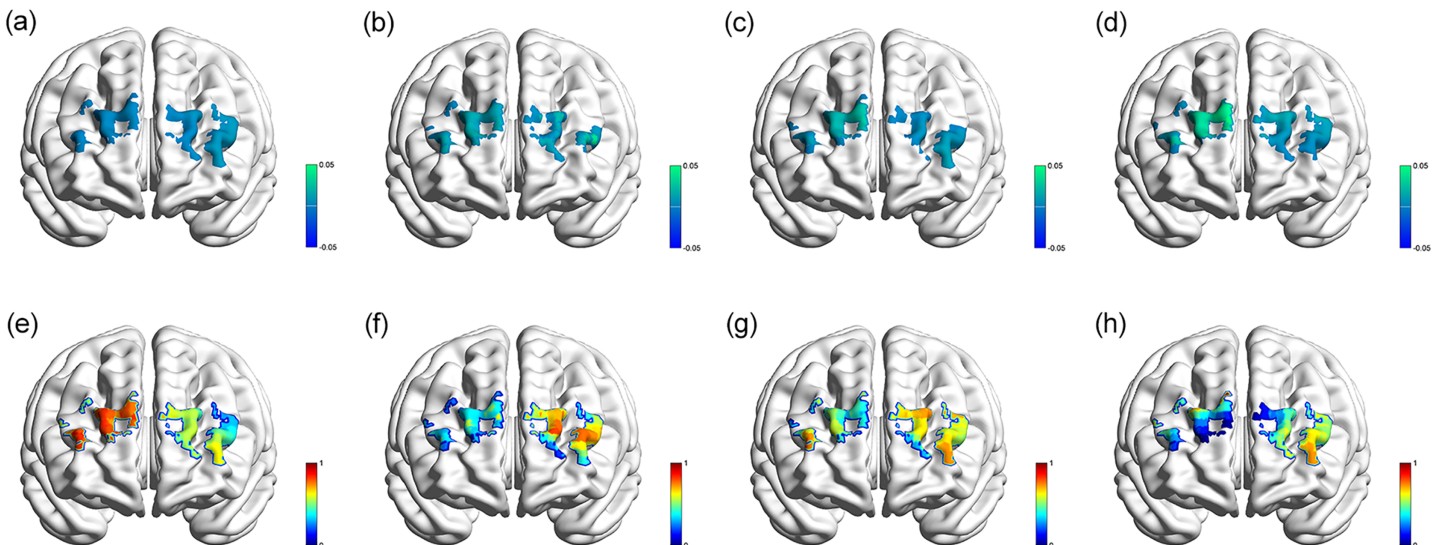

**Figure 2 Topographic images.** β-map of the prefrontal activations derived from oxyHb, showing associations between channel activation and accuracy of the incongruent Stroop tests in regular sleep duration (A), light short sleep (B), mild short sleep (C), severe short sleep (D) groups; p-map of the prefrontal activations derived from oxyHb, showing associations between channel activation and accuracy of the incongruent Stroop tests in regular sleep duration (E), light short sleep (F), mild short sleep (G), severe short sleep (H) groups.

With regard to the regular sleep group, a marginal significant negative correlation between the activation of left middle frontal gyri (Ch. 17) and accuracy of the incongruent Stroop tests (β 95% CI: −0.009 [−0.018 to 0.001], $p = 0.063$) was observed. In the light short sleep group, this study found a positive activation correlation between oxyHb levels in specific Ch. 19 (β 95% CI: 0.024 [0.004–0.045], $p = 0.026$) and 20 (β 95% CI: 0.023 [0.007–0.039], $p = 0.011$), which represented the left dorsolateral superior frontal gyri. In the mild short sleep group, a similar result was identified and a marginal significant negative correlation between the activation of the left dorsolateral superior frontal gyri (Ch. 20) and accuracy of the incongruent Stroop test was observed, with β 95% CI: −0.012 [−0.024 to 0.001], $p = 0.068$. In the severe short sleep group, oxyHb relocated to more regions of the PFC. The activation of the following channels was significantly associated with accuracy of the incongruent Stroop test, including Ch. 5 (right middle frontal gyri, β 95% CI: 0.014 [0.005–0.023], $p = 0.005$), Ch. 8 and 11 (right dorsolateral superior frontal gyri, β 95% CI: −0.017 [−0.030 to −0.004], $p = 0.016$; β 95% CI: −0.014 [−0.027 to −0.001], $p = 0.035$), Ch.12 (right superior and middle frontal gyri, β 95% CI: 0.05 [0.013–0.088], $p = 0.012$), and Ch. 17 (left middle frontal gyri, β 95% CI: −0.012 [−0.022 to −0.003], $p = 0.010$).

Figure 2 demonstrates the topographic views (β-map and p-map) of correlations mentioned above. The results of the prefrontal cortex activation in each group of regular sleep duration, light short sleep, mild short sleep, and severe short sleep have been provided in Tables S3, S4, S5, S6, providing detailed results of the incongruent Stroop tests. In this study, changes of oxyHb in more regions of the PFC were activated in severe short sleep.

## DISCUSSIONS

This current research explored the association between sleep duration and cognitive performance in college students. Stratified analysis was conducted under different subgroups. Further hemodynamic mechanisms were detected using fNIRS equipment. Thereto, we assessed sleep duration *via* the ActiGraph GT3X+ accelerometer and recorded prefrontal cortical hemodynamics during the cognitive test (Stroop test) by fNIRS. This study confirmed that short sleep was associated with decreased cognitive performance represented by lower accuracy during the incongruent Stroop tests.

Short sleep was associated with decreased Stroop accuracy. In accordance with previous evidence, substantial findings showed that sever short sleep (*i.e.*, sleep deprivation) might increase the risk of error accidents, with literature demonstrating that one-time extreme sleep deprivation was related to decline of individuals' executive behavior similarly to intoxicated alcohol users (*Correa, Molina & Sanabria, 2014*) . One meta-analysis focusing on the influence of acute sleep deprivation on cognition reported significant differences for both reaction speed and accuracy (*Lim & Dinges, 2010*). Recent neuroimaging studies indicated that chronic sleep deprivation might induce structural changes in the brain, affecting areas associated with learning and memory (*Nofzinger, 2004*; *Voldsbekk et al., 2021*). Additionally, functional MRI studies have shown altered patterns of brain activation during cognitive tasks following sleep deprivation (*Hsu et al., 2023*; *Helakari et al., 2023*).

In contrast to forced sleep deprivation studies, this study examined the cognitive performance of college students who were naturally short-sleepers. Similar to our research, a cross-sectional study in medical students found that a lack of sleep significantly influenced their academic performance (*Barahona-Correa et al., 2018*). Another study investigated the relationship between sleep patterns and academic performance in undergraduates, and further confirmed that poor academic performance was related to delayed sleep timing (*Phillips et al., 2017*). Although the relationship between sleep and academic outcomes has been examined in a certain number of studies, this study provided additional insights into the hazard of short sleep on cognitive tests by Stroop tasks, which re-emphasized the importance of ensuring sleep time for cognitive performance in young adults, especially for female groups. To mitigate the influence of short sleep, there was evidence that physical exercise would be beneficial to improve cognitive function (*You et al., 2023b*).

This current study also assessed the changing mechanisms from the perspective of hemodynamics using fNIRS. Cognitive processes were active processes involved in a number of brain circuitry while sleep may function to support the reorganization of related neurons (*Tononi & Cirelli, 2014*). This study demonstrated that in the severe short sleep group, the right dorsolateral superior frontal gyri was negatively activated. It was also found that sleep-deprived participants showed altered cerebral blood flow in the right middle cerebral artery (*Csipo et al., 2021*). There were other studies using the functional connection of fNIRS to investigate the impact of sleep loss on cognitive performance. One research reported that decreased sustained attention caused by high sleep pressure might

lead to impaired connectivity in the left PFC (*Borragan et al., 2019*; *Mukli et al., 2021*). A brain physiological explanation for these regions' negative activation was related to the theory of hemodynamics (*Shmuel et al., 2002*). According to this interpretation, nearby areas of the brain would have a decline in cerebral blood flow when local cerebral blood flow increased within the brain. Observation of more channels activated in the severe short sleep group showed that lack of sleep might be associated with cognitive fatigue. Therefore, more resources in the PFC were needed to maintain cognitive function.

To the best of our knowledge, this was the first study to investigate the association between sleep duration and cognitive performance in college students using fNIRS equipment. As opposed to most previous cross-sectional studies that used self-reported interviews to collect sleep information, this study applied an accelerometer as a wearable tracker to objectively assess sleep duration. Given that previous research has reported the influence of acute sleep deprivation on cognitive function, it seemed that this study was promising to explore the relationship between measures of regular short sleep and cognitive performance. Further studies should explore diverse neuroscientific dimensions, including neural networks, neurotransmitter systems, and the basal ganglion, as alterations in these dimensions may exert distinct impacts on cognitive functions. Moreover, in contrast to more research focusing on clinical patients or extreme sleep deprivation interventions, it was also recommended that studies should be conducted to investigate the effects of nature short sleep in a cohort of healthy populations with a larger sample size for broader applicability.

Though the findings of this study were interesting, several limitations needed to be acknowledged. Firstly, due to the cross-sectional design, it was unable for us to infer causality from these results. Secondly, using accelerometers alone could not differentiate between different sleep stages, such as REM and NREM sleep. Employing polysomnography (PSG) could provide more comprehensive sleep stage information, enhancing the assessment of sleep influence. Thirdly, even when we considered the potential mechanisms using fNIRS, only the PFC was chosen as the region of interest. Recent neurophotonic guidelines put forward a strict data acquisition process, which will provide more reporting items and details (*Novi et al., 2020*; *Yucel et al., 2021*). It is crucial to acknowledge this limitation imposed by the finite number of channels available in this study. Due to this constraint, we were unable to simultaneously cover all the specific brain areas of interest. While we did not claim to fully elucidate all underlying mechanisms, this study intended to uncover significant hemodynamic associations between sleep duration and cognitive abilities. Future studies should examine other brain areas using fMRI and EEG to assess systemic physiological changes of the brain in depth. Last but not least, influencing factors such as physical activity level and food intake were not considered, and further randomized controlled trials are warranted to verify study findings.

## CONCLUSIONS

The findings of this study supported the idea that sleep had a crucial role to play in cognitive functioning in young adults and females were more vulnerable to short sleep

durations. Hemodynamic changes should be considered as crucial contributors of the mechanism. Scientific advances in this study centered on enhancing the understanding of the PFC's activation that connect sleep duration with cognitive performance. This knowledge will not only enrich the current literature on sleep-cognition interactions but also provide a foundation for future studies to explore potential interventions or strategies aimed at optimizing cognitive function in young adults through sleep management. Although this study cannot be used to make causal inferences regarding the relationships between sleep duration and cognitive performance, our results nevertheless highlighted that obtaining enough sleep was associated with better cognitive performance. Further experimentation is necessary to identify factors that predispose individuals to adopt short sleep and explore the underlying molecular mechanisms.

### Funding

This study was supported by the Institute of Sports Development Research of Tsinghua University (Research on John Mo's thought and practice of Physical Education) and the China Postdoctoral Science Foundation (2022M711858). The funders had no role in study design, data collection and analysis, decision to publish, or preparation of the manuscript.

### Grant Disclosures

The following grant information was disclosed by the authors:
Institute of Sports Development Research of Tsinghua University.
China Postdoctoral Science Foundation: 2022M711858.

### Competing Interests

The authors declare that they have no competing interests.

### Author Contributions

- Yanwei You conceived and designed the experiments, performed the experiments, analyzed the data, prepared figures and/or tables, authored or reviewed drafts of the article, and approved the final draft.
- Jianxiu Liu conceived and designed the experiments, performed the experiments, analyzed the data, prepared figures and/or tables, authored or reviewed drafts of the article, and approved the final draft.
- Xingtian Li performed the experiments, analyzed the data, authored or reviewed drafts of the article, and approved the final draft.
- Peng Wang performed the experiments, analyzed the data, authored or reviewed drafts of the article, and approved the final draft.
- Ruidong Liu conceived and designed the experiments, authored or reviewed drafts of the article, and approved the final draft.
- Xindong Ma conceived and designed the experiments, authored or reviewed drafts of the article, and approved the final draft.

## Human Ethics

The following information was supplied relating to ethical approvals (*i.e.*, approving body and any reference numbers):

Academic ethics committee of Tsinghua University

## Data Availability

The raw data is available in the Supplemental File.

## Supplemental Information

Supplemental information for this article can be found online at http://dx.doi.org/10.7717/peerj.17057#supplemental-information.

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
