# Peer review of "Relationship between accelerometer-measured sleep duration and Stroop performance: a functional near-infrared spectroscopy study among young adults"

_PeerJ, doi:10.7717/peerj.17057_

## Round 0.1 · original submission · Major Revisions

I have thoroughly reviewed the manuscript and concur with Reviewer 1's perspective. I believe it would greatly benefit from a more defined focus, specifically elaborating on the methods used to achieve clarity. Additionally, within the Discussion section, emphasizing the manuscript's core objectives and what it aims to elucidate would be advantageous.

Finally, I apologize for the length of the peer review process.

**Language Note:** The review process has identified that the English language must be improved. PeerJ can provide language editing services - please contact us at copyediting@peerj.com for pricing (be sure to provide your manuscript number and title). Alternatively, you should make your own arrangements to improve the language quality and provide details in your response letter. – PeerJ Staff

Reviewer 1 ·

Basic reporting

English used is not so clear and unambiguous in some places.
Article structure is ok, references proposed cover sufficiently the field but they need to be expand.

Experimental design

`Research question is defined but important details for the scientific background are missing.
Methods (analysis) have to be described more.

Validity of the findings

The section should highlight the novelty of the research and focus on answering the original research questions without providing excessive speculation and perspectives. The conclusions require further refinement.

Additional comments

The relationship between accelerometer-measured sleep duration and Stroop performance in young adults: a functional near-infrared spectroscopy study

As the title suggests, this exploratory study aims to provide information on the relationship between accelerometer-measured sleep duration and cognitive performance on the Stroop test in young adults, and the underlying haemodynamic mechanisms as measured by fNIRS. Although the topic of the study is interesting, the current version of the manuscript lacks many details and some sections (discussion) are not relevant enough. The English language should be improved throughout the manuscript to ensure that we can clearly understand the text.

Introduction
This section needs more background in order to provide a stronger rationale for the present study, in case it appears exploratory. Overall, more detail is strongly required to better justify your study.
L40. Please indicate the duration (range) of short sleep based on the current literature for young adults.
L44-45. Can the authors give more information on what type of cognitive function? Please be more specific about "poor cognitive (loss of %?; which cognitive functions? Add some details here) and academic (to be precise) performance".
L46-47. Do you mean a correlation? If so, what is the degree of association?
L47. Neural mechanisms. This part should be expanded given the aim of the present study.
L48. Perspective on haemodynamics is proposed, but at this stage the authors fail to give relevant findings on neural mechanisms (probably based on electrophysiological studies, with EEG method) and to argue why haemodynamic patterns need to be studied.
L54-55. Again, please give more information on this study (23). What is the role of neurovascular coupling in the cognitive process?
L60. This information "susceptibility to the effects of short sleep" is fundamental to your justification. Authors must provide details of references 26 and 27 in order to better justify their choice (PFC, fNIRS?), while indicating the gap they wish to fill. Novelty must be addressed here.
L62. "Current evidence" for fNIRS and sensor-based accelerometers needs to be added before (see previous comment for fNIRS). For sensor-based accelerometers, please give some reasons for proposing this measure (monitoring sleep duration/quality, I suppose?).
L63-64. The authors did not propose some hypotheses. This could be explained by the exploratory nature of this study. However, some of the literature is not detailed enough (17, 18, 23, 26, 27). Please adapt accordingly.

Methods used
L82. Why did you choose this threshold of 3 days per week? 3 out of 7 days seems low. Why do you say that the data are reliable? How was it assessed?
L84. The analysis of the actigraph sensors should be detailed. What measures (and what about their sensitivity) were used?
L84. When was the Stroop test administered? This is unclear.
Please rephrase the last sentence "their cortex" is not appropriate here. Add participants""

Discussion (without s L 216)

L230-236. Please explain what is the added value of this study compared to yours and others dealing with the same issues.
L256-260. This section is out of scope.
L265. fNIRS can measure other cortical regions of interest.
L269. An item of fNIRS analysis needs to be added to the Methodological considerations or limitations subsection. Because current guidelines (see Neurophotonics guidelines published in the last two years) for fNIRS have not been strictly applied.
L272. The results did not show that the haemodynamic changes were the main contributors to the mechanism (of what, by the way?). The present study shows haemodynamic correlates of sleep duration.
L281. It is surprising to suggest this perspective. Please remove it. It is outside the scope of the study.

·

Basic reporting

Authors of this work have attempted to study the correlation between sleep deprivation and cognitive performance. The study is interesting, however, the manuscript needs some major revisions.

The topic and abstract were not pointed. Authors probably resorted to using complex technical vocabulary with an expression such as "...accelerometer-measured sleep duration and Stroop performance..." when it could have simply been "...relationship between sleep deprivation and cognitive performance using ..."

The presentation of the topic and abstract made it quite difficult for the study objective to be understood. Somewhere in the abstract, the relationship between cognitive function and the Stroop test should have been expressly stated.

Experimental design

Authors should provide a graphical abstract that summarizes the study methodology in one figure fore easier comprehension.

Validity of the findings

Lines 23-37 of the abstract should be rewritten for easier comprehension. For example: "A total of 58 participants were included in this study" belongs to the methods and not results.

"Results showed that less sleep duration was associated with lower accuracy of the incongruent Stroop test." What does this mean? Why is the meaning of this statement not written in the abstract?

The discussion was expressed like a theoretical repetition of the results and had no explanation on the causative rationales behind the results. For example lines 230-231 states "a cross-sectional study in medical students found that a lack of sleep significantly influenced their academic performance..." The question is WHY? Almost everyone, everywhere knows this. But the novel question from the study is why does lack of sleep affect academic performance? Does it shrink the brain? Does it cause some tissue damage? What is the biological mechanism behind such results?

Additional comments

The introduction was too focused on basic details about what lack of sleep can cause. Almost everyone in the world knows these facts! A technical study as this should explain, in comprehensive terms, what the novel approach was in solving the problem, and how the terminologies of the expected findings matter.

The use of pronouns "we" and "our" is not too ethical for a technical paper. This should be revised in the entire manuscript.

Reviewer 3 ·

Basic reporting

The study effectively outlines the objectives and methods used to investigate the relationship between sleep duration, cognitive performance, and hemodynamic changes. Overall, the study contributes valuable insights into the relationship between sleep duration, cognitive function, and hemodynamics in young adults, highlighting the significance of maintaining sufficient sleep for cognitive performance.

Experimental design

The study was well designed, even though expanding th sample size could provide a broader generalization of the results.

Validity of the findings

The findings regarding the association between shorter sleep duration and decreased accuracy in the incongruent Stroop test are compelling. However, more detailed discussion and interpretation of the observed hemodynamic changes and their specific implications for cognitive performance would add depth to the results. Additionally, discussing limitations, such as potential confounding variables or other factors influencing sleep quality, would provide a more comprehensive understanding.

The utilization of ActiGraph GT3X+ accelerometers for sleep duration assessment and 24-channel fNIRS for hemodynamic measurements represents cutting-edge methodology. However, discussing potential limitations or validations of these measurement tools in the context of this specific study would fortify the methodological rigor.

---

## Round 0.2 · Minor Revisions

Your revised manuscript is now suitable for further publication. But several points were raised by a reviewer. Please correct these points according to the comments from a reviewer.

Reviewer 1 ·

Basic reporting

This new version of the article has been greatly improved in a number of ways.
The introduction is now clear and concise, while highlighting the contribution of the present study and its originality. The limitations of this pioneering study are clearly indicated.

However, two sentences should be deleted.
L268-269. The following sentence is too speculative and irrelevant (the cognitive fatigue was not evaluated here): "This indicated less sleep may induce cognitive fatigue and relocating blood hemodynamics in more brain areas".
L20-21. This sentence should also be deleted from the summary: "This indicated less sleep may induce cognitive fatigue and relocating blood hemodynamics in more brain areas."

Experimental design

no comment

Validity of the findings

no comment

·

Basic reporting

The manuscript seems to have been improved. It might be suitable for acceptance.

Experimental design

Optimal corrections seem to have been made.

Validity of the findings

Proper detailing of statistical analysis for results seem to have been improved.

---

## Round 0.3 · accepted · Accept

The editor confirmed that the authors have appropriately addressed the reviewers' comments and that the manuscript is worthy of publication.
Congratulations!